# Oxidative Stress Response and *E. coli* Biofilm Formation under the Effect of Pristine and Modified Carbon Nanotubes

**DOI:** 10.3390/microorganisms11051221

**Published:** 2023-05-06

**Authors:** Yuliya Maksimova, Anastasiya Zorina, Larisa Nesterova

**Affiliations:** 1Laboratory of Molecular Biotechnology, Institute of Ecology and Genetics of Microorganisms UB RAS, 614081 Perm, Russia; cjcyf_92@mail.ru; 2Department of Microbiology and Immunology, Perm State University, 614990 Perm, Russia; 3Laboratory of Adaptation of Microorganisms, Institute of Ecology and Genetics of Microorganisms UB RAS, 614081 Perm, Russia; larisa.nesterova@bk.ru; 4Department of Plant Physiology and Soil Ecology, Perm State University, 614990 Perm, Russia

**Keywords:** carbon nanotubes, oxidative stress, reactive oxygen species, biofilms, extracellular polymeric substances

## Abstract

The article investigates the expression of *oxyR* and *soxS* oxidative stress genes in *E. coli* under the effect of pristine multi-walled carbon nanotubes (MWCNTs) and pristine single-walled carbon nanotubes (SWCNTs), MWCNTs and SWCNTs functionalized with carboxyl groups (MWCNTs-COOH and SWCNTs-COOH, respectively), SWCNTs functionalized with amino groups (SWCNTs-NH_2_) and SWCNTs functionalized with octadecylamine (SWCNTs-ODA). Significant differences were found in the expression of the *soxS* gene, while no changes were observed in the expression level of the *oxyR* gene. The pro-oxidant effect of SWCNTs, SWCNTs-COOH, SWCNTs-NH_2_, and SWCNTs-ODA is presented, and the contrary antioxidant effect of pristine MWCNTs and MWCNTs-COOH in the presence of methyl viologen hydrate (paraquat) is shown. The article shows that SWCNTs-COOH, SWCNTs-NH_2_, and SWCNTs-ODA added to the medium generate the production of reactive oxygen species (ROS) in bacterial cells. SWCNTs-COOH intensified the *E. coli* biofilm formation, and the biofilm biomass exceeded the control by 25 times. Additionally, it is shown that the *rpoS* expression increased in response to MWCNTs-COOH and SWCNTs-COOH, and the effect of SWCNTs-COOH was more significant. SWCNTs-COOH and SWCNTs-NH_2_ initiated an increase in ATP concentration in the planktonic cells and a decrease in the biofilm cells. The atomic force microscopy (AFM) method showed that the volume of *E. coli* planktonic cells after the exposure to carbon nanotubes (CNTs) decreased compared to that without exposure, mainly due to a decrease in cell height. The absence of a strong damaging effect of functionalized SWCNTs on *E. coli* K12 cells, both in suspension and in biofilms, is shown. Contact with functionalized SWCNTs initiated the aggregation of the polymeric substances of the biofilms; however, the cells did not lyse. Among the studied CNTs, SWCNTs-COOH caused an increased expression of the *soxS* and *rpoS*, the formation of ROS, and stimulation of the biofilm formation.

## 1. Introduction

Carbon nanotubes (CNTs) are innovative materials whose unique characteristics have led to their active use in electronics, instrumentation, and composite materials [1]. They also have great prospects in medicine, agriculture, biotechnology, and other industries [2,3,4]. Functionalized CNTs are a support for enzyme immobilization and are used in the nanobiocatalysis [5]. Additionally, CNTs in the composition of hybrid nanomaterials have a great potential for use in the field of catalysis, optoelectronics, and biomedicine [6]. There are single-walled carbon nanotubes (SWCNTs), multi-walled carbon nanotubes (MWCNTs), CNTs exhibiting the properties of metals or non-metals, pristine CNTs, and CNTs functionalized with chemical groups [7,8]. 

The cytotoxic effects of CNTs on prokaryotes and eukaryotes were noted [9]. The first information about the antibacterial action of CNTs concerned the effect of SWCNTs on *E. coli* [10,11]. However, the mechanism of the antibacterial effect of CNTs remains a subject of debate. One of the main mechanisms is a direct disruption of the microbial cell integument, followed by release of the cytoplasmic and cell lysis. In this case, CNTs were compared with “nanodarts” [12]. The negative effect of CNTs on the cell can be associated with: (1) membrane disruption, or its oxidation; (2) oxidative stress caused by the formation of ROS; (3) toxicity of impurities; and (4) bacterial agglomeration [13].

The effects of CNTs can vary from a strong cytotoxic effect to a positive effect. It depends on the type of CNTs (SWCNTs or MWCNTs, pristine or functionalized, purified or containing impurities, exhibiting the properties of metals or non-metals, having different lengths and diameters, etc.) and the microbial object itself (prokaryotes or eukaryotes, Gram-negative or Gram-positive bacteria, plankton or biofilm). For example, SWCNTs have been shown to cause membrane damage [11] and affect *E. coli* biofilms [14]. However, the atomic force microscopy (AFM) method showed that the lysis of *E. coli* cells is caused not by pure SWCNTs, but by metal impurities in SWCNTs [15]. It has been shown that MWCNTs damage bacterial cells to a lesser extent than SWCNTs [11]. As shown by Arias and Yang, MWCNTs do not exhibit antimicrobial activity against either Gram-negative or Gram-positive bacterial cells, while SWCNTs exhibit strong antimicrobial activity against both cell types [16]. Functionalization enhances the antibacterial action of MWCNTs [17]. Microbial biofilms are less affected by nanomaterials, mainly due to the protective effect of the exopolysaccharide matrix [14]. The effect of CNTs on biofilm formation and the mature biofilms depends on the type of CNTs and the bacteria themselves. For example, it has been established that MWCNTs have a greater effect on the biofilm biomass of Gram-negative rather than Gram-positive bacteria; the effect depends on functionalization and is strain specific [18,19].

The antimicrobial effect of nanoparticles, including CNTs, can be caused by oxidative stress [20]. Chen and Jafvert reported that pristine and functionalized CNTs can exhibit photo reactivity initiating the formation of reactive oxygen species (ROS) and/or their transformation in the natural environment [21]. SWCNTs functionalized with carboxyl groups in the form of aqueous colloidal dispersions generated ROS, namely, singlet oxygen, superoxide anion, and hydroxyl radicals under oxygen conditions in the light (λ = 300–410 nm) [21]. The main oxidative defense regulons in *E. coli* are the oxyR regulon (*katG* gene), which responds to an increase in the intracellular concentration of hydrogen peroxide, and the soxRS regulon (*soxS* gene), whose expression increases with changes in redox conditions in the cell and is activated by superoxide anions [22,23]. Quite a lot of research is devoted to oxidative stress that occurs in eukaryotic cells under the influence of CNTs [24]. Different studies have demonstrated the ability of MWCNTs to induce ROS in different cell lines [25,26]. Kang et al. reported that oxidative stress is a possible mechanism that partially explains the toxicity of CNTs to bacteria. They found that several genes that are expressed after cell exposure to SWCNTs and MWCNTs, are part of the soxRS and oxyR systems associated with the bacterial oxidative stress response [11]. However, studies relating to the induction of soxRS and oxyR regulons have not been conducted. In addition, the data obtained are quite contradictory: both pro-oxidant [27] and antioxidant effects of CNTs [28] on bacteria have been revealed. The expression of *oxyR*, *soxS*, and *rpoS* in genetically modified *E. coli* strains harboring oxyR′::lacZ, soxS′::lacZ, and rpoS::lacZ under the effect of carbon nanomaterials has not been studied previously. The first comprehensive study of expression of these genes, ROS formation, ATP content and morphological changes in *E. coli* planktonic and biofilm cells under the effect of CNTs is presented. The SWCNT-ODA effect on bacteria has been studied for the first time. The objective of the work is to study the pro-oxidant and antioxidant effects of various CNTs on *E. coli* and to establish the effect of CNTs on the ROS formation, biofilm formation, ATP content, and morphological changes in planktonic and biofilm cells.

## 2. Materials and Methods

### 2.1. The Strains and Culture Conditions

The expression of *oxyR* and *soxS* oxidative stress genes was studied on genetically modified strains of *E. coli* BGF 930 (RK4936 λ[Φ(oxyR′::lacZ)]), BGF 940 (MC4100 λ[Φ(oxyR′::lacZ)]) [29] and *E. coli* EH40 (GC4468 λEH40 (soxS′::lacZ) [30], kindly provided by Bruce Demple (Stony Brook University, Stony Brook, NY, USA). The *rpoS* gene expression was studied on *E. coli* RO91 (MC4100 λRZ5:rpoS742::lacZ[hybr]), kindly provided by Regina Hengge (University of Berlin, Berlin, Germany). The strains were cultivated in Luria-Bertani (LB) medium (VWR, Radnor, PA, USA).

Biofilm formation was studied on *E. coli* K12 ATCC. The strain was cultivated in LB medium (DiaM, Moscow, Russia).

### 2.2. Carbon Nanotubes

In the present study, pristine and functionalized MWCNTs and SWCNTs were used. The pristine MWCNTs (“Taunit-M”, “NanoTechCenter” Ltd., Tambov, Russia) had external diameter of 10–30 nm, internal diameter of 5–15 nm, length ≥ 2 µm, specific surface area ≥ 270 m^2^ g^−1^, and bulk density of 0.025–0.06 g cm^−3^. The functionalized MWCNTs (“Taunit-M”, “NanoTechCenter” Ltd., Tambov, Russia) had the external diameter of 20–50 nm, internal diameter of 10–20 nm, length ≥ 2 µm, and specific surface area ≥ 160 m^2^ g^−1^. The carboxylated MWCNTs (MWCNTs-COOH) had 0.1–1.0 mmol/g COOH groups. The pristine SWCNTs (“TUBALL^TM^”, “OCSiAl”, Novosibirsk, Russia) had average outer diameter of 1.6 ± 0.4 nm, total specific surface area of 1311 m^2^ g^−1^, and purity of 99.5%. The modified SWCNTs (Sigma-Aldrich, St. Loius, MO, USA) had the various functional groups. The carboxylated SWCNTs (SWCNTs-COOH) had more than 90% carbon basis, length of 0.5–1.5 µm, and diameter of 4–5 nm; the aminated SWCNTs (SWCNTs-NH_2_) had more than 90% carbon basis, length of 0.7–1.0 µm, and diameter of 4–6 nm; the SWCNTs functionalized with octadecylamine (SWCNTs-ODA) had 80–90% carbon basis, length of 0.5–2.0 µm, and diameter of 2–10 nm. 

### 2.3. Chemical List

Chemicals: methyl viologen hydrate (paraquat) (Aldrich, Schnelldorf, Germany), o-nitrophenyl-β-D-galktopyranoside (Sigma, Cibolo, TX, USA), 2′7′-dichlorofluorescein diacetate (Sigma, Jerusalem, Israel), crystal violet (Reachim, Moscow, Russia), ATP Bioluminescent Assay Kit (Sigma, USA), and BacLight^TM^ Bacterial Viability Kits (Invitrogen, Waltham, MA, USA) were used in the work.

### 2.4. Gene Expression

*E. coli* BGF 930 (RK4936 λ[Φ(oxyR′::lacZ)]), *E. coli* BGF 940 (MC4100 λ[Φ(oxyR′::lacZ)]), *E. coli* EH40 (GC4468 λEH40 (soxS′::lacZ)), and *E. coli* RO91 (MC4100 λRZ5:rpoS742::lacZ[hybr]) were cultivated for 7–8 h at 37 °C in LB broth (VWR, USA). Then, 50 μL of inoculum was added to 50 mL of LB broth and cultivated for 15 h in Erlenmeyer flasks at 100 rpm in a thermostated shaker GFL 1092 (GFL, Berlin, Germany). The culture was diluted with LB broth to OD600 = 0.1 in 40 mL Erlenmeyer flasks. After the culture reached OD600 = 0.3, it was transferred to Erlenmeyer flasks containing 10 mL of LB broth and 200 mg/L CNTs. The mixture was preliminarily treated with ultrasound in the Elma Ultrasonic 30S bath (Elma, Singen, Germany) at 37 kHz 10 times for 1 min. Paraquat (PQ) at a concentration of 1 µg/mL was added. In all cases, the nutrient medium contained 25 µg/mL streptomycin. To study the antioxidant effect of CNTs, PQ was added together with CNTs.

The expression of the *oxyR*, *soxS*, and *rpoS* genes was assessed by the activity of β-galactosidase. The activity of β-galactosidase was assessed by the Miller method based on the ability of this enzyme to hydrolyze o-nitrophenyl-β-D-galktopyranoside (ONPG). After sampling, 200 µL of the culture was placed in test tubes in 1800 µL of Z-buffer (Na_2_HPO_4_—0.06 M; NaH_2_PO_4_—0.04 M; KCl—0.01 M; MgSO_4_—0.001 M; β-mercaptoethanol—0.05 M; pH 7.0) and treated with a mixture of 0.1% sodium dodecyl sulfate (20 µL) and chloroform (40 µL), with vigorous shaking for 10 s. The reaction was started with 400 µL ONPG (4 mg/mL). The reaction was carried out for 15 min at 28 °C and stopped by adding 1 mL of 1 M Na_2_CO_3_ to the reaction mixture. Before measurement, 3 mL of distilled water was added to the test tubes. The color intensity was assessed with a UV 1280 spectrophotometer (Shimadzu, Kyoto, Japan) by optical density (OD420).

The calculation of CFU was carried out after inoculation of successive tenfold dilution of the culture into a LB solid nutrient medium.

### 2.5. Assessment of ROS

The intracellular level of ROS was assessed by increasing the fluorescence of 2′7′-dichlorofluorescein diacetate, which penetrated cells, de-esterified and oxidized intracellularly to highly fluorescent 2′7′-dichlorofluorescein [31]. The level of 2′7′-dichlorofluorescein diacetate oxidation in *E. coli* K12 cells was assessed in the presence of 200 mg/L of CNTs. *E. coli* K12 was grown for 24 h, centrifuged, and resuspended to a final density of 10^6^ CFU/mL in 0.9% NaCl with the addition of SWCNTs-COOH, SWCNTs-NH_2_, and SWCNTs-ODA; 95 µL of this mixture was dropped into the each well of black flat-bottomed 96-well plates (Nunc, Rosklide, Denmark). *E. coli* K12 suspension with PQ (1 and 10 µg/mL), CNTs without *E. coli* K12, and *E. coli* K12 without CNTs were used as controls. Then, 5 µL of 2′7′-dichlorofluorescein diacetate was added to all wells and thermostated for 30 min at 37 °C. Fluorescence was measured at λ 485/530 (ex/em) by an Infinite M1000 Pro plate reader (Tecan, Männedorf, Switzerland) every hour for 9 h in total.

### 2.6. Assessing Biomass Biofilm

*E. coli* K12 biofilms were grown for 24 h in the wells of a 96-well plate in 200 µL of LB medium inoculated with 5 µL of bacterial suspension containing (1.5–1.7 × 10^9^) CFU/mL. The LB medium contained 200 mg/L of CNTs, and the LB medium without CNTs served as a control. To obtain a homogeneous suspension, the medium with CNTs was preliminarily treated with ultrasound in the Elma Ultrasonic 30S bath, Elma (Germany) at 37 kHz 10 times for 1 min. Planktonic cells were removed from the wells by decantation, the biofilm was washed twice with 200 µL of potassium phosphate buffer, and the biofilm biomass was assessed. The biofilm was stained with 0.1% crystal violet for 40 min in the dark. Then, the staining agent was removed, the stained biofilm was washed once with potassium phosphate buffer, and the staining agent was extracted with 96% ethanol (200 µL). Biofilm formation was assessed by the optical density of the staining solution at 540 nm using an Infinite M1000 Pro plate reader (Tecan, Männedorf, Switzerland).

### 2.7. Assessing ATP Content

*E. coli* K12 biofilms were grown for 24 h at 30 °C in LB medium in 96-well flat-bottomed plates in 200 µL of LB medium with 200 mg/L SWCNTs-COOH and SWCNTs-NH_2_ inoculated with 5 µL of a bacterial suspension containing (1.5–1.7 × 10^9^) CFU/mL. *E. coli* K12 biofilms grown without CNTs were controls. The biofilms were washed twice with potassium phosphate buffer, and 200 µL of DMSO was added. After 15 min exposure to DMSO, the samples were frozen at −18 °C. A suspension of *E. coli* K12 was grown in a conical flask on LB broth with CNTs (200 mg/L) for 24 h, centrifuged, washed from the medium with potassium phosphate buffer, and resuspended in 0.9% NaCl to OD540 = 1. Then, 2 mL of suspension was placed in the test tubes of Eppendorf type and then centrifuged; the supernatant was removed, and the cells were destroyed by adding 200 μL of DMSO. In the second variant, *E. coli* K12 was grown in LB broth without additives, as described earlier, the biofilms were washed with potassium phosphate buffer, and 200 μL of 0.9% NaCl containing 200 mg/L of SWCNTs-COOH and SWCNTs-NH_2_ was added. *E. coli* K12 biofilms with 0.9% NaCl without CNTs were controls. Then, 200 µL of 0.9% NaCl with CNTs was also added to the precipitate of suspension grown on LB medium without additives and washed as described above. After 2 h of exposure, the liquid was removed from the biofilms. The suspension culture was centrifuged, and the supernatant was removed. After that, 200 µL of DMSO was added to biofilms and cells’ precipitate.

A kit of reagents (ATP Bioluminescent Assay Kit) was used to determine the ATP concentration. The samples were diluted 10 times with deionized water, and 100 µL of the sample was mixed with 100 µL of the reagent containing luciferin and firefly luciferase. Luminescence intensity was measured on an Infinite M1000 Pro plate reader (Tecan, Männedorf, Switzerland). The amount of ATP was determined according to the calibration curve, and for biofilms, it was calculated per well of the plate and for planktonic cells per 1 mL of suspension (1 mL contained 3.4 mg of dry cells).

### 2.8. Combined Atomic Force Microscopy and Confocal Laser Microscopy

The morphology of bacterial cells and surface profiles was studied using a combined system of AFM Asylum MFP-3D-BIO (Asylum Research, Santa Barbara, CA, USA) and a confocal laser scanning microscope (CLSM) Olympus FV1000 (Olympus Corporation, Tokyo, Japan) in the laboratory of atomic force and confocal microscopy at the Rhodococcus-Center of Perm State University. Cell viability was assessed after obtaining CLSM images, for which preparations were preliminarily stained with Live/Dead^®^ (Syto9/propidium iodide) BacLight^TM^ Bacterial Viability Kits and incubated for 15 min in the dark. Cell surface profiles were studied using AFM. Scanning was performed in the semi-contact mode in air using OMCL-AC240TS-R3 silicon cantilevers (Olympus, Taichung, Taiwan) coated with aluminum, with a resonant frequency of 70 (50–90) kHz, a needle curvature radius of 7 nm, and a stiffness constant of 2 (0.6–3.5) N/m. To determine the linear dimensions of cells (length, width, height) and to characterize the structure of the cell surface (roughness, Sq,), we obtained two- and three-dimensional topographic images of bacteria. The shape of the cells was taken as an ellipsoid, the volume of the cells was calculated by the formula:V = 4/3π(a/2)(b/2)(c/2), µm^3^,(1)
where a is the length, µm, b is the width, µm, and c is the height, µm.

Preparations for AFM scanning were prepared as follows. *E. coli* K12 biofilms were grown for 72 h in LB broth on 24 × 50 mm glasses (Deltalab, Barcelona, Spain), washed with 0.9% NaCl, and placed for 1 h in 0.9% NaCl containing 200 mg/L of CNTs. The liquid was removed, and the samples were stained with Live/Dead^®^ for 20 min, dried, and scanned. The cell suspension was centrifuged for 10 min at 14,000× *g*, washed with 0.9% NaCl, centrifuged again, and 0.9% NaCl containing 200 mg/L CNTs was added to cells’ precipitate. The control was 0.9% NaCl without CNTs. After 1 h of exposure, the suspension was centrifuged, and the supernatant was removed from the CNTs, and then cells’ precipitate was diluted in 0.9% NaCl with Live/Dead^®^, applied to a scanning glass, stained for 20 min, and dried. The microphotographs were processed using the Igor Pro 6.22A program (WaveMetrics, Portland, OR, USA).

### 2.9. Statistical Analysis

The presented data are the results of three independent experiments. The results obtained were processed statistically, and the means, standard deviations, and confidence intervals were determined. The significance of differences was assessed using Student’s *t*-test, *p* < 0.05. The means and the standard deviations are presented in the Appendix A.

## 3. Results

### 3.1. Expression of the oxyR and soxS Genes

The expression of *oxyR* and *soxS* genes included in the oxidative stress protection regulons of OxyR and SoxRS was evaluated in the presence of carbon nanomaterials. There was no change in the level of expression of the *oxyR*; however, significant differences were found in the expression of the *soxS* (*p* < 0.05). The expression of the *soxS* was increased for exposure to pristine and functionalized SWCNTs (Figure 1). When exposed to SWCNTs, SWCNTs-COOH, SWCNTs-NH_2_, and SWCNTs-ODA, the difference from the control (bacteria suspension without CNTs) was significant. Under the effect of MWCNTs “Taunit-M”, the level of *soxS* expression was significantly lower than in the control. The number of CFU/mL was controlled, and it was shown that the number of viable cells increased during growth for 4.5 h, increasing from 0.8–1.7 × 10^8^ to 6.1–8.0 × 10^8^, and neither CNTs nor PQ had an inhibitory or lethal effect on *E. coli*.

The antioxidant effect of carbon nanomaterials was assessed in the presence of a strong oxidizing agent (PQ). It was found that MWCNTs “Taunit-M” and MWCNTs-COOH added to PQ had a pronounced antioxidant effect. For these nanomaterials, the difference from the control (PQ) was significant (Figure 2).

Thus, for the first time, the pro-oxidant effect of SWCNTs, SWCNTs-COOH, SWCNTs-ODA, and SWCNTs-NH_2_ and, conversely, the antioxidant effect of pristine MWCNTs and MWCNTs-COOH in the presence of PQ have been shown.

### 3.2. ROS Production

The production of ROS was estimated by the increase in the fluorescence of membrane-permeable 2′7′-dichlorofluorescein diacetate, de-esterified, and oxidized intracellularly to highly fluorescent 2′7′-dichlorofluorescein for 9 h. The level of oxidation of 2′7′-dichlorofluorescein diacetate in *E. coli* K12 cells was assessed in the presence of SWCNTs-COOH, SWCNTs-NH_2_, and SWCNTs-ODA, which showed a significant pro-oxidant effect on *E. coli*. It was shown that SWCNTs-COOH, SWCNTs-NH_2_, and SWCNTs-ODA added to the medium generated ROS in bacterial cells (Figure 3). The highest level of ROS was observed under the effect of SWCNTs-NH_2_. These nanotubes also initiated the oxidation of the reagent in the absence of cells; however, the fluorescence level in this case was lower than in the case of ROS formation in the presence of cells. The smallest production of ROS was observed under the effect of SWCNTs-ODA.

### 3.3. Biofilm Formation of E. coli K12 under the Effect of Functionalized SWCNTs

The biofilm biomass in LB broth under the effect of SWCNTs was assessed. It has been shown that SWCNTs-COOH caused an increase in the *E. coli* biofilm formation. The biofilm biomass exceeds the control by 25 times (Figure 4a). SWCNTs-NH_2_ and SWCNTs-ODA, which also showed a pro-oxidant effect, did not increase biofilm biomass.

After the destruction of *E. coli* K12 biofilms for 24 h in 0.9% NaCl, the largest biofilms biomass also remained after exposure to SWCNTs-COOH (Figure 4b). It exceeded the biofilms biomass after exposure to 0.9% NaCl without CNTs by 3.4 times.

### 3.4. Expression of the rpoS

The expression of *rpoS* under the effect of MWCNTs-COOH and SWCNTs-COOH was determined (Figure 5). The expression of *rpoS* was the most pronounced in the phase of culture growth deceleration from 2.5 to 9 h. It was shown that both MWCNTs-COOH and SWCNTs-COOH cause an increase in the expression of the *rpoS* compared to the control without exposure. The increase in the expression of this gene under the effect of SWCNTs-COOH was more significant. The number of CFU/mL increased within 10 h from (1.37 ± 0.32) × 10^8^ to (2.37 ± 0.24) × 10^9^, (2.51 ± 0.44) × 10^9^, and (3.10 ± 0.39) × 10^9^ in the control, in the medium with MWCNTs-COOH and that with SWCNTs-COOH, respectively.

### 3.5. ATP Content

The ATP content in the *E. coli* K12 biofilm and planktonic cells was assessed during 24 h growth in LB broth with SWCNTs-COOH and SWCNTs-NH_2_, and during the 2 h effect of these CNTs on the grown suspension and mature biofilms. It was shown that the ATP content significantly increases in the planktonic cells and significantly decreases in the biofilm cells under the effect of SWCNTs-COOH (Figure 6a).

When exposed to SWCNTs-NH_2_ (Figure 6b), the same trend is observed, but the ATP content changes significantly only when nanotubes act on the mature biofilms and grown planktonic culture.

### 3.6. Morphology of E. coli K12 Planktonic and Biofilm Cells after Exposure to CNTs

After exposure for 1 h to functionalized SWCNTs, which showed a pro-oxidant effect to *E. coli*, the morphological changes in *E. coli* K12 planktonic and biofilm cells were studied, and their morphometric parameters (length, width, height of cells, total cell volume, and surface roughness) were determined. Cells were preliminarily stained with Live/Dead^®^ fluorescent dye (Syto 9/propidium iodide) BacLight^TM^ Bacterial Viability Kit, and morphometric parameters were determined separately for living and dead cells. The minimum number of living cells was found upon exposure to SWCNTs-NH_2_ in a suspension of *E. coli* K12; however, significantly more cells with an intact membrane were in the biofilm. Table 1 shows the dimensions and roughness of cells with permeable membranes that were red after staining with Live/Dead^®^. The volume of cells and the roughness of their surface was shown to be less in the biofilm cells than those of the planktonic cells (Table 1). The roughness of cells changed insignificantly upon exposure to CNTs, except for the impact of SWCNTs-ODA. These nanotubes led to a significant increase in the surface roughness of both planktonic and biofilm cells.

The volume of planktonic cells after exposure to CNTs decreased compared to that without exposure, mainly due to a decrease in cell height. When exposed to SWCNTs-ODA, the length and width of the cell slightly decreased, but not the height. No such effect was found on biofilm cells.

The morphology of *E. coli* K12 after one hour of exposure to functionalized SWCNTs was studied. It was shown that cells undergo the greatest morphological changes when exposed to SWCNTs-NH_2_ and SWCNTs-ODA (Figure 7). A significant part of the biofilm cells was covered with rounded formations, which was presumably a polymer matrix aggregated with nanotubes.

Thus, the AFM-CLSM method showed the absence of a strong damaging effect of functionalized SWCNTs on *E. coli* K12 cells both in suspension and in biofilms. Clear outlines of cells confirm the absence of lysis. The vast majority of non-viable cells were found only in samples exposed to SWCNTs-NH_2_.

## 4. Discussion

We have studied the effects of pristine SWCNTs and MWCNTs, MWCNTs-COOH, SWCNTs-COOH, SWCNTs-NH_2_, and SWCNTs-ODA on *E. coli* cells. For SWCNTs-COOH, SWCNTs-NH_2_, and SWCNTs-ODA, which showed a pro-oxidant effect, the formation of ROS was confirmed. The biofilm formation of *E. coli* under SWCNTs-COOH, SWCNTs-NH_2_, and SWCNTs-ODA and the effect of these carbon nanomaterials on the mature biofilms, ATP content, morphometric parameters and cell morphology in plankton and biofilm were studied. The -COOH and NH_2_ groups interact with amino acids and proteins, forming chemical bonds. As a result, CNTs functionalized with these groups have a greater affinity for the cell surface than pristine ones. Octadecylamine is a functional group of SWCNTs-ODA. ODA has the properties of primary aliphatic amines, is toxic to bacteria, and is used for the preservation of metal surfaces. In this regard, the antibacterial effect of nanotubes functionalized with ODA groups was studied.

The antibacterial effect of CNTs depends on various factors. Among these, not only are the properties of CNTs themselves (their diameter, length, functionalization, and others) important, but also the properties of the environment in which they operate. CNTs did not inhibit the growth of *E. coli* in LB broth, and CFU/mL after 5 h of growth was equal to that in the control. However, when *E. coli* cells were exposed to CNTs for 1 h in 0.9% NaCl, a significant number of cells with damaged membranes were detected, as evidenced by the red color of the cells when stained with the Live/Dead^®^ dye. In this case, different effects could be associated with different degrees of CNT aggregation. In the LB broth, CNT aggregation is more significant than in the mineral medium. The damaging effect of well-dispersed nanoparticles on cells is known to be higher than aggregated ones. Kang et al. suggested that the toxicity of MWCNTs for *E. coli*, which increases with fineness, is associated with an increase in the effective surface area of nanotubes in contact with bacterial cells [32]. The more CNTs are dispersed in the medium, the more kinetic energy they have, which allows them to penetrate cells at a high rate [33].

Surface modification of CNTs affects their toxicity because the surface reactivity and their ability to aggregate with bacteria are changed. Hydrophobic unmodified CNTs are poorly dispersed, and they agglomerate and precipitate faster than those functionalized with hydrophilic groups [13]. Arias and Yang note that neutral or negatively charged SWCNTs-OH and SWCNTs-COOH aggregate with bacteria more efficiently and reduce their viability, in contrast to positively charged SWCNTs-NH_2_ [16]. Our studies have shown that SWCNTs-NH_2_ also have a damaging effect on bacterial cells, and ROS production under the effect of SWCNTs-NH_2_ is higher than that of SWCNTs-COOH. SWCNTs, unlike MWCNTs, create a closer contact with the bacterial cell wall, thus damaging it. The hypothetical mechanisms of the effect of CNTs on animal or human cell lines (which may also be applicable to microorganisms) are as follows: increased membrane permeability, toxicity of metal impurities, physical puncture, suppression of energy metabolism, inhibition of enzyme activity, and induced oxidative stress [16]. CNTs produce excess ROS, causing the degradation of cell membranes, adenosine triphosphate, and deoxyribonucleic acid, leading to severe cell damage. In this case, a three-stage mechanism of cytotoxicity is postulated: (1) ROS generation on the CNT surface, (2) oxidative stress, and (3) membrane destruction and cell death [34].

It has been noted that oxidative stress is one of the mechanisms by which carbon nanomaterials affect bacterial cells [11,13,27,35]. Oxidative stress can be formed due to the oxidative-generating properties of the particles themselves, as well as their ability to stimulate the production of ROS in cells. In the first case, the sources of oxidative stress are transition metal impurities used as catalysts in the production of non-metallic nanoparticles, including SWCNTs. In the second case, these are relatively stable free radical intermediates that are present on the surfaces of particles or redox groups formed as a result of the functionalization of nanoparticles [24].

Martín et al. report that exposure for 24 h to 100 mg/L SWCNTs and MWCNTs induces oxidative stress in the cells of the pathogenic yeast *Candida albicans* and the bacteria *Staphylococcus aureus* and *Pseudomonas aeruginosa* by generating ROS [27]. The mechanism of SWCNTs toxicity was investigated using SWCNTs-mediated oxidation of glutathione, which is a mediator of the redox state of the cell. An increase in the ratio of metallic SWCNTs accompanied an increase in the degree of glutathione oxidation. Scanning electron microscopy images of *E. coli* demonstrated morphological changes in cells consistent with the results of cytotoxicity and glutathione oxidation. Cell membranes had increased roughness and integrity violations after contact with metal SWCNTs [32]. In our study, the AFM method showed no major damage to the integrity of *E. coli* planktonic and biofilm cells in contact with functionalized SWCNTs. Lysed cells were not detected, although these types of SWCNTs caused oxidative stress.

The present study shows that SWCNTs-COOH, SWCNTs-NH_2_, and SWCNTs-ODA increase the level of expression of *soxS*, but do not affect *oxyR*. The *oxyR* gene is required for the induction of oxyR regulon, which includes hydrogen peroxide-inducible genes. It was shown that SWCNTs-COOH generated singlet oxygen, superoxide anion, and hydroxyl radicals under oxygen conditions in the light [21]. Our experiments with 2′7′-dichlorofluorescein diacetate demonstrated that SWCNTs-COOH, SWCNTs-NH_2_, and SWCNTs-ODA generate ROS. It is assumed that functionalized nanotubes can act as electron donors or participate in the transfer of electrons from other donors; in the latter case, NADH may be involved [21]. Since there are no changes in the expression of the *oxyR*, there is reason to believe that these carbon nanomaterials do not initiate the formation of hydrogen peroxide in the *E. coli* cells.

Pristine MWCNTs, as well as MWCNTs-COOH, did not show any pro-oxidant effect. MWCNTs and MWCNTs-COOH exhibited an antioxidant effect in the presence of PQ, as judged by a decrease in the *soxS* expression level. There are studies that show that purified MWCNTs do not generate free radicals but are effective scavengers of ROS. Fenoglio et al. show that MWCNTs are effective scavengers of hydroxyl radicals and superoxide anions, although the exact molecular mechanism of this phenomenon is unknown [28]. Li et al. report that MWCNTs increases the activity of five antioxidant enzymes in grape seedlings under salt stress [36]. On the contrary, Tan et al. show that during cultivation in suspension with MWCNTs in rice cells, the ROS content increases, and the cell viability decreases [37]. When antioxidants are introduced into the medium, the ROS content and viability return to the initial level. The following mechanism for ROS formation in cells is proposed: at the outer surface of the plasma membrane, it is mediated by NADPH oxidases, while at the cell wall matrix, it is associated with the effect of class I peroxidases, poly(di)amine oxidases, and oxalate oxidase. MWCNTs interact with the cell wall, possibly activating these enzymes, leading to the formation of ROS [37]. Rajavel et al. noted that CNTs are toxic to living cells due to their ability to initiate lipid peroxidation, which leads to membrane damage. The pristine CNTs were effective radical generators, while CNTs functionalized with tannic and gallic acid antioxidants did not produce them and were able to neutralize free radicals formed in vitro and slow down lipid peroxidation [38].

Data on the antioxidant and pro-oxidant effects of CNTs are rather contradictory. There is an opinion that more hydrophilic graphite surfaces are less able to destroy the lipid membrane of bacteria than hydrophobic ones; therefore, functionalized (i.e., with reduced hydrophobicity) CNTs have less effect on cell viability. It has been shown that functionalized CNTs are capable of both generating higher ROS and quenching or removing free radicals [33]. Our study shows for the first time that not only MWCNTs but also MWCNTs-COOH reduce the effect of PQ on *E. coli* cells, while functionalized SWCNTs, including the -COOH group, cause oxidative stress. Consequently, the pro-oxidant or antioxidant effect depends not so much on the functional group as on the type of CNTs (MWCNTs or SWCNTs effecting the cell). It is known that SWCNTs are more damaging to bacterial cells than MWCNTs. Our data suggest that oxidative stress is not caused by CNTs themselves, but it is a consequence of membrane damage and uncoupling of the respiratory chain, which leads to an excess of free radicals in the cell.

Biofilm formation is a natural bacteria lifestyle. It is known that more than 90% of prokaryotes exist in the form of biofilms. The planktonic form for the majority of bacteria is a way of settling in the environment when biofilms are destroyed due to maturation, starvation, and the action of shear forces [39]. In addition, the formation of a biofilm is considered as an adaptive response reaction of microorganisms. At the level of gene expression, the process of biofilm formation is quite complex and depends on many factors. The formation of bacterial biofilm is mainly dependent on two key regulatory factors, namely, the Quorum Sensing (QS) system and bis-(3′-5′)-cyclic dimeric guanosine monophosphate (c-di-GMP), which is a second messenger and enhances the production of extracellular polysaccharides [40,41]. As a result of stressful situations, such as starvation, bacterial cells reduce the amount of c-di-GMP by activating phosphodiesterase, which promotes the expansion of biofilm cells [42]. It is known that it is the exopolysaccharide matrix that protects biofilm cells from the effects of adverse environmental factors. In this case, the interaction of CNTs with the biofilm begins at the extracellular polymeric substances (EPS) level. Polysaccharides produce highly viscous aqueous solutions that can interfere with nanoparticle transport and interaction with cells [14]. It was shown that sublethal doses of UV irradiation inhibited biofilm growth by reducing the number of bacteria, reducing the thickness of the biofilm, and disrupting the barrier effect. Surviving bacteria demonstrated strong oxidative stress and EPS secretion in response to this stress at the initial stage of sublethal photocatalysis. The expression of genes associated with EPS secretion, biofilm growth, and antibiotic resistance increased. This resulted in an increase in biofilm thickness and protection against photocatalysis [43]. Thus, biofilm formation may be indirectly associated with oxidative stress.

The protective role of the exopolysaccharide matrix is well known [44,45]. In the case of the impact of nanoparticles on microbial biofilms, the matrix acts as the first level of protection, and the effect on cells depends on the diffusion of particles in the matrix. The diffusion of nanoparticles in a biofilm increases with decreasing nanoparticles size and biofilm density [46]. Using the AFM method, we showed the presence of round formations on the surface of *E. coli* K12 biofilm cells after exposure to functionalized SWCNTs. The outlines of the cells are clear, which indicates the absence of lysis. It has been suggested that rounded formations are EPS aggregated with CNTs. In this case, the biofilm cells are protected from the action of functionalized SWCNTs, since their penetrating ability decreases with aggregation.

The study shows that the *E. coli* biofilm biomass significantly increases in the presence of SWCNTs-COOH in the medium. However, the ATP concentration is significantly lower than in cells without exposure. This may be due to the fact that *E. coli* actively forms a biofilm in response to a stress factor (the presence of nanotubes in the medium). However, the mature biofilm contains a significant number of dead cells. A similar fact was observed by Alizadeh et al. They showed that *E. coli* and *S. aureus* grew on composite membranes coated with MWCNTs-COOH, but less than 4% of living cells remained [47]. According to our data, in the planktonic cells, on the contrary, the content of ATP increased. It may be associated with stress, which results in the uncoupling of energy and constructive metabolism. When exposed to SWCNTs-NH_2_, the same trend was noted, but the change in the ATP concentration was significant only when exposed to the grown culture and biofilms.

It is known that oxyR regulon activity decreases with an increase in rpoS activity, which reduces unnecessary energy expenditure with low nutrient levels and growth retardation. In addition, tight control of these overlapping regulons prevents over-regulation of shared genes, since a number of genes are co-regulated by OxyR and RpoS. Partial modulation of regulon expression occurs at the level of regulators, where RpoS and OxyR, as well as SoxS, control each other’s expression. For example, OxyR negatively regulates *rpoS* translation, while for the soxRS regulon, RpoS is a positive regulator of *soxS* expression [48]. RpoS is also known to play a role in biofilm formation. During the formation of *E. coli* biofilms, it has been shown that rpoS activation promotes biofilm maturation by activating genes involved in adhesion and stress response and suppressing genes involved in flagellar synthesis and energy metabolism [49]. Since we noted a significant increase in biofilm biomass in response to SWCNTs-COOH, the effect of SWCNTs-COOH on *rpoS* expression was studied in the next experiment. MWCNTs-COOH was taken as a comparison, which did not cause an increase in the expression of *soxS* gene and did not lead to increased biofilm formation. The expression of *rpoS* increased both in response to SWCNTs-COOH and MWCNTs-COOH, but for SWCNTs-COOH, it was more significant. Our study shows that the expression of soxRS and rpoS regulon’s genes, the content of ROS in the cell, and the biofilm formation are enhanced under the effect of SWCNTs-COOH. These processes were the most pronounced for this type of CNTs.

Thus, CNTs, even the most damaging to the cell, such as functionalized SWCNTs, are unlikely to be highly effective antibacterial agents, since they do not the cause the death of the entire population. At the same time, SWCNTs-COOH generating oxidative stress, also lead to increased biofilm formation. While the total ATP content in biofilms decreases compared to the control, it remains at the level of 10^−10^ M/well, which indicates the presence of living cells in the biofilm.

Challenges and future perspectives. Prospects for further work include testing composite materials with various CNTs and assessing biofilm formation on such surfaces. The antibacterial action can be induced by the surface roughness and the physical or chemical effects, and it can be achieved by functionalizing, derivatizing, or polymerizing the surfaces [50]. Such studies will make it possible to create materials with different properties, either antifouling or supporting biotechnologically significant biofilms.

## 5. Conclusions

The study revealed that SWCNTs-COOH, SWCNTs-NH_2_, and SWCNTs-ODA initiated a significant increase in the level of expression of the soxRS regulon protecting against oxidative stress but not *oxyR*, in *E. coli*. The study shows that oxidative stress is caused by the formation of ROS in the cells under the effect of these CNTs. At the same time, MWCNTs-COOH and pristine MWCNTs have an antioxidant effect. SWCNTs-COOH promote *E. coli* biofilm formation, leading to a 25-fold increase in biofilm biomass compared to the control. The ATP content in *E. coli* K12 biofilm cells grown with SWCNTs-COOH was 1.12 × 10^−10^ M versus the control, 2.94 × 10^−10^ M, and in planktonic cells it was 3.01 × 10^−10^ M versus the control 2.33 × 10^−10^ M. SWCNTs-COOH and MWCNTs-COOH increased the expression of *rpoS*; however, its level was higher when exposed to SWCNTs-COOH. At the same time, the presence of CNTs in a rich nutrient medium did not lead to a decrease in the number of CFUs during several hours of growth. Treatment with CNTs did not initiate a lysis of *E. coli* K12 planktonic and biofilm cells. The vast majority of cells with a permeable membrane were found only in the samples exposed to SWCNTs-NH_2_. The roughness of the cell surface significantly increased under the effect of SWCNTs-ODA and amounted to 126.21 ± 10.12 nm, versus 90.19 ± 7.75 nm in the control. The volume of planktonic cells decreased by a factor of 1.3–1.4 times after exposure to SWCNTs-NH_2_ and SWCNTs-COOH due to a decrease in cell height. Biofilm cells were covered with rounded formations, which may have appeared during the aggregation of the polymeric matrix with nanotubes. Thus, functionalized SWCNTs did not cause an immediate lethal effect and lysis of *E. coli* cells; however, they generated an oxidative stress response. The level of expression of the *soxS* gene was significantly lower than when exposed to the oxidizing agent (PQ) but significantly higher than in untreated cells.

Thus, we showed that functionalized and pristine SWCNTs cause oxidative stress in *E. coli* cells through the formation of ROS, while functionalized and pristine MWCNTs act as antioxidants. The enhancement of biofilm formation in *E. coli* K12 under the effect of SWCNTs-COOH is the more significant. The exopolysaccharide matrix of biofilms plays a protective role with respect to CNTs effect. EPS initiate CNTs aggregation, which decreases their destructive effect.

## Figures and Tables

**Figure 1 microorganisms-11-01221-f001:**
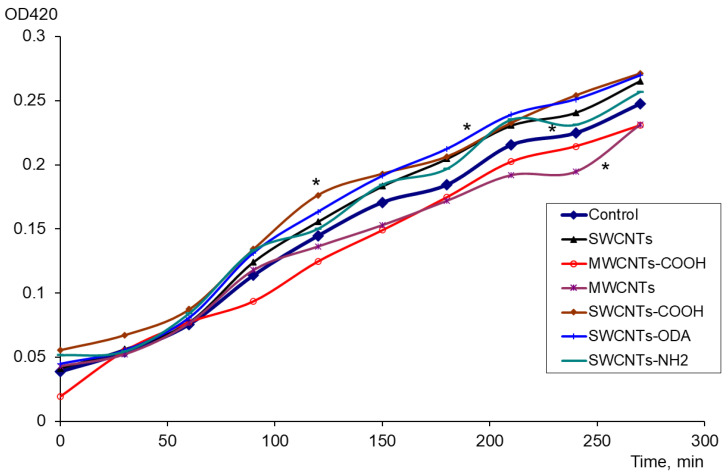
The effect of CNTs on the *soxS* expression: changes in the activity of β-galactosidase (OD420) in control, with PQ and CNTs, * *p* < 0.05. Statistical analysis is presented in the Appendix A.

**Figure 2 microorganisms-11-01221-f002:**
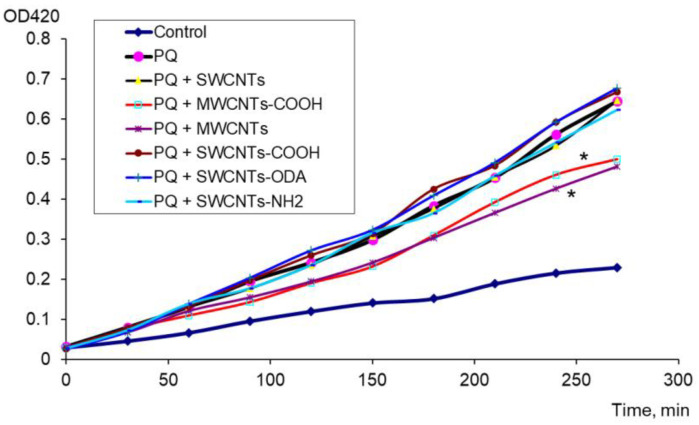
Antioxidant effect of CNTs in the presence of PQ, * *p* < 0.05. Statistical analysis is presented in the Appendix A.

**Figure 3 microorganisms-11-01221-f003:**
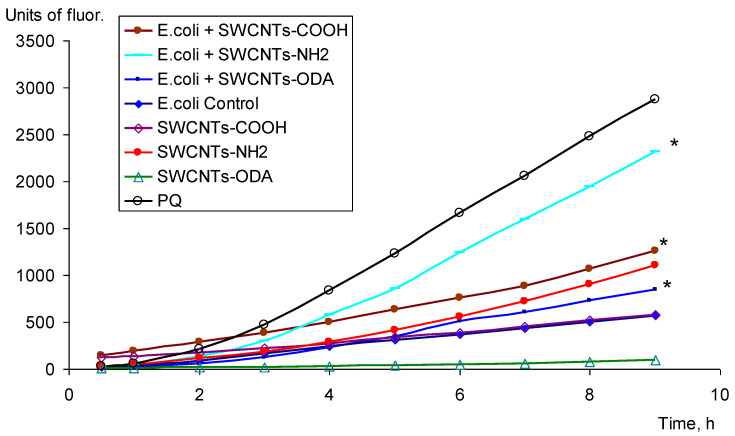
ROS production under CNTs, * *p* < 0.05. Statistical analysis is presented in the Appendix A.

**Figure 4 microorganisms-11-01221-f004:**
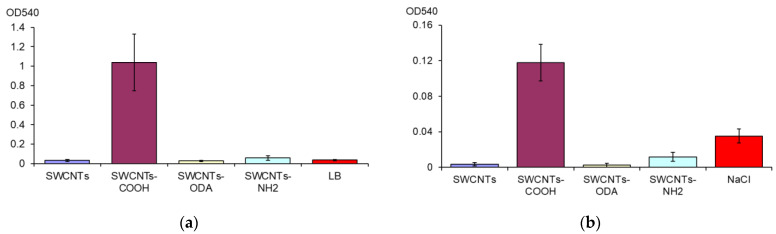
Biofilm biomass of *E. coli* K12 in LB medium in the presence of CNTs (**a**) and after destruction for 24 h in 0.9% NaCl with CNTs (**b**).

**Figure 5 microorganisms-11-01221-f005:**
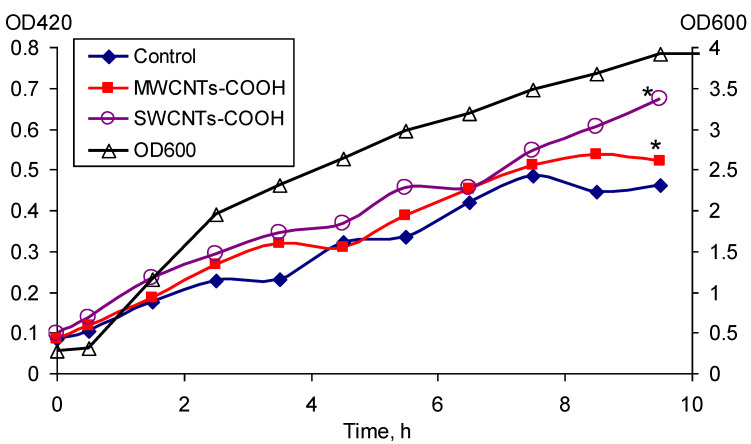
Expression of the *rpoS* gene under the effect of MWCNTs-COOH and SWCNTs-COOH, * *p* < 0.05. Statistical analysis is presented in the Appendix A.

**Figure 6 microorganisms-11-01221-f006:**
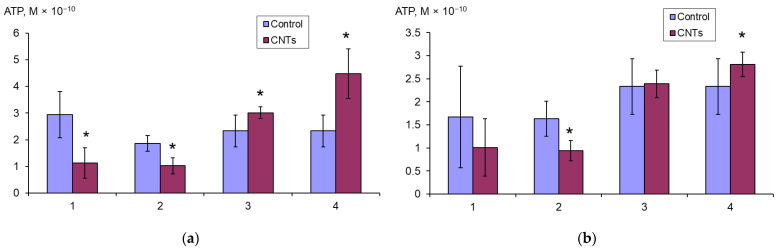
ATP content (M × 10^−10^) in *E. coli* K12 biofilms (1, 2) calculated per well, and planktonic cells (3, 4) calculated per mL, after 24 h of growth (1, 3) with SWCNTs-COOH (**a**) and SWCNTs-NH_2_ (**b**), and after 2 h exposure to mature biofilms (2) and to grown planktonic culture (4), * *p* < 0.05.

**Figure 7 microorganisms-11-01221-f007:**
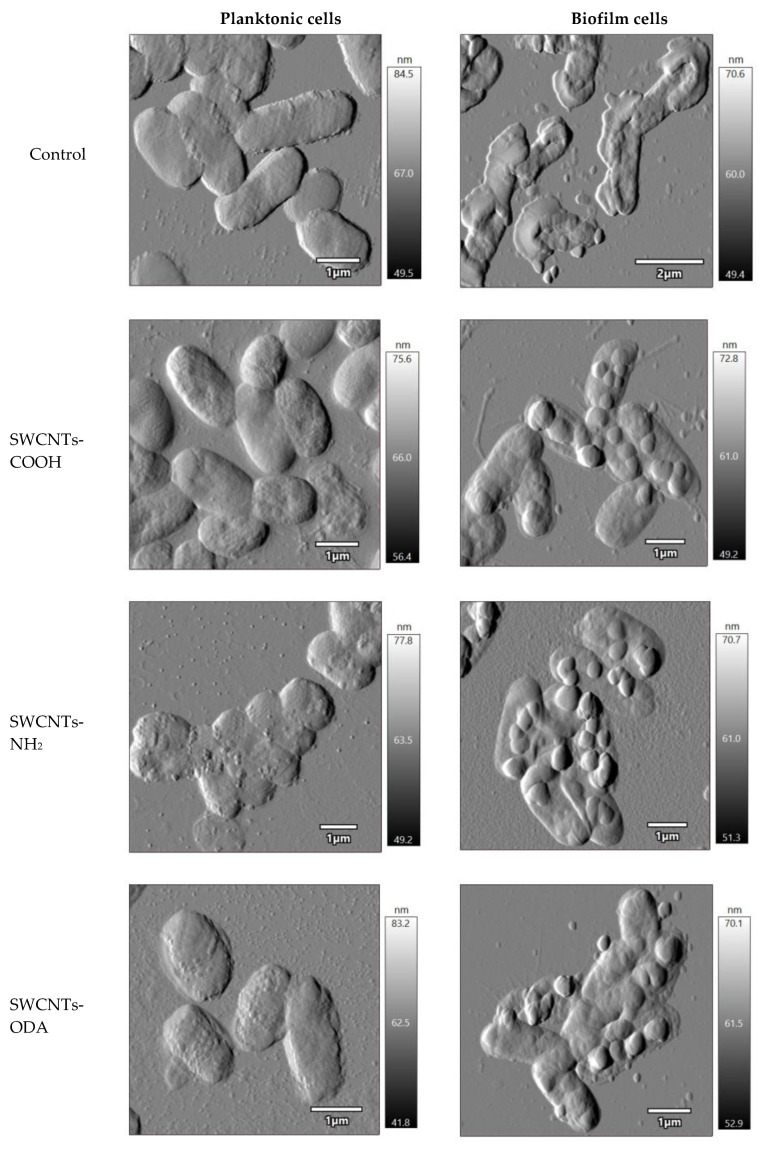
Morphology of *E. coli* K12 planktonic and biofilm cells after exposure to functionalized SWCNTs.

**Table 1 microorganisms-11-01221-t001:** Morphometric parameters of *E. coli* K12 planktonic and biofilm cells after exposure to functionalized SWCNTs.

CNTs	Roughness, nm		Length, µm		Width, µm		Height, µm		Cell Volume, µm^3^	
	1	2	1	2	1	2	1	2	1	2
Control	90.19 ± 7.75	85.17 ± 7.425	2.24 ± 0.10	2.71 ± 0.22	1.16 ± 0.02	1.08 ± 0.05	0.27 ± 0.01	0.16 ± 0.01	0.37	0.24
SWCNTs-COOH	92.43 ± 8.51	69.53 ± 4.93	2.27 ± 0.12	2.54 ± 0.15	1.14 ± 0.03	0.99 ± 0.05	0.21 ± 0.01	0.16 ± 0.01	0.28	0.21
SWCNTs-NH_2_	88.51 ± 13.82	82.00 ± 10.30	2.65 ± 0.24	1.98 ± 0.14	0.98 ± 0.03	1.07 ± 0.06	0.19 ± 0.01	0.19 ± 0.01	0.26	0.20
SWCNTs-ODA	126.21 ± 10.12	91.00 ± 7.80	2.00 ± 0.10	2.18 ± 0.17	1.10 ± 0.04	1.00 ± 0.04	0.30 ± 0.01	0.17 ± 0.02	0.34	0.20

Note: 1—planktonic cells; 2—biofilm cells.

## Data Availability

The data presented in this study are available in Appendix A.

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
