# Peer review of "Oxidative Stress Response and E. coli Biofilm Formation under the Effect of Pristine and Modified Carbon Nanotubes"

_microorganisms, 2023, doi:10.3390/microorganisms11051221_

Round 1

Reviewer 1 Report

Greetings, Editor thank you for providing me with the opportunity to review the article. I reviewed the article with ID = microorganisms-2342316. Overall, the article structure and content are suitable for the Microorganisms (ISSN 2076-2607) journal. I am pleased to send you major comments. Please consider these suggestions as listed below.         

  1. The title seems very ok.
  2. Abstract also seems ok.
  3. Key words seems weird. It should be short and not more than 5. Please revise it.
  4. Research gap should be delivered on more clear way with directed necessity for the future research work.
  5. Introduction section must be written on more quality way, i.e., more up-to-date references addressed.
  6. The novelty of the review must be clearly addressed and discussed, compare previous research with existing research findings and highlight novelty.
  7. The reference 1-3 need to delete and simply cite this single reference here- Yaqoob, A.A.; Ibrahim, M.N.M.; Ahmad, A.; Vijaya Bhaskar Reddy, A. Toxicology and Environmental Application of Carbon Nanocomposite. In Environmental Remediation through Carbon Based Nano Composites; Jawaid, M., Ahmad, A., Ismail, N., Rafatullah, M., Eds.; Springer Nature: Singapore, 2021; pp. 1–18.
  8. What is the main challenge of field?
  9. Several sentences in introduction are very long. Please rephrased it. The long sentences are not ideal to understand it. Please revise your paper accordingly since some issue occurs on several spots in the paper.
  10. Please check the abbreviations of words throughout the article. All should be consistent.
  11. What is problem statement?
  12. The main objective of the work must be written on the more clear and more concise way at the end of introduction section.
  13. Please provide space between number and units. Please revise your paper accordingly since some issue occurs on several spots in the paper.
  14. In section 2 please provide all chemical list with specification in a separate .
  15. So far the result and discussion is well explained but the comparison with literature is missing.
  16. Please add a comparative profile section.
  17. Regarding the replications, authors confirmed that replications of experiment were carried out. However, these results are not shown in the manuscript, how many replicated were carried out by experiment? Results seem to be related to a unique experiment. Please, clarify whether the results of this document are from a single experiment or from an average resulting from replications. If replicated were carried out, the use of average data is required as well as the standard deviation in the results and figures shown throughout the manuscript. In case of showing only one replicate explain why only one is shown and include the standard deviations.
  18. Please add a section about challenges and future perspectives.
  19. Conclusion section is missing some perspective related to the future research work, quantify main research findings, and highlight relevance of the work with respect to the field aspect.
  20. To avoid grammar and linguistic mistakes, MAJOR level English language should be thoroughly checked. Please revise your paper accordingly since several language issue occurs on several spots in the paper. Its quite suprising, article is from English-country but still there are several grammatical mistakes.
  21. Reference formatting need carefully revision. All must be consistent in one format. Please follow the journal guidelines.

Author Response

Dear Reviewer,

Thank you for your review. We highly appreciate your helpful comments on our manuscript. We also very much appreciate your suggestions, which have been very helpful in improving the manuscript. We tried to take into account your comments and make the necessary corrections.

  1. Key words are shortened.
  2. We indicated a research gap at the end of the introduction: “However, studies related to the induction of soxRS and oxyR regulons have not been conducted. In addition, the data obtained are quite contradictory: both pro-oxidant [25] and antioxidant effects of CNTs [26] on bacteria have been revealed. Expression of oxyR, soxS and rpoS in genetically modified E. coli strains harboring oxyR′::lacZ, soxS'::lacZ and rpoS::lacZ under the effect of carbon nanomaterials has not been studied previously. The study of the CNTs effect on bacterial cells will make it possible to choose the types of CNTs for different purposes (antibacterial material or supports for biofilms which are important for biotechnologies)."
  3. In the introduction, we used modern references (2020-2022) and references to pioneering works in the field of studying the antibacterial action of CNTs (2007-2008, Kang et al., Elimelech et al.) We added 2 more references to modern articles. We have been reformulated a number of sentences. There are 28 references in the introduction, 10 are not older than 5 years and all the rest are not older than 2005.
  4. We presented the novelty of the study is more clearly: "Expression of oxyR, soxS and rpoS in genetically modified E. coli strains harboring oxyR′::lacZ, soxS'::lacZ and rpoS::lacZ under effect of carbon nanomaterials has not been studied previously. The first comprehensive study of these genes' expression, ROS formation, ATP content and morphological changes of E. coli planktonic and biofilm cells under the effect of CNTs is presented." The novelty of our study is presented in comparison with known studies on this topic: "Kang et al. reported that oxidative stress is a possible mechanism that partially explains the toxicity of CNTs to bacteria. They found that several genes that are expressed after cell exposure to SWCNTs and MWCNTs, are part of the soxRS and oxyR systems associated with the bacterial oxidative stress response [9]. However, studies related to the induction of soxRS and oxyR regulons have not been conducted. In addition, the data obtained are quite contradictory: both pro-oxidant [25] and antioxidant effects of CNTs [26] on bacteria have been revealed."
  5. We have included the reference proposed by the reviewer under the number 1. Yaqoob, A.A.; Ibrahim, M.N.M.; Ahmad, A.; Vijaya Bhaskar Reddy, A. Toxicology and Environmental Application of Carbon Nanocomposite. In Environmental Remediation through Carbon Based Nano Composites; Jawaid, M., Ahmad, A., Ismail, N., Rafatullah, M., Eds.; Springer Nature: Singapore, 2021; pp. 1–18. In our opinion, this reference does not fully cover the statement "They also have great prospects in medicine, agriculture, biotechnology and other industries". Therefore, we would like to leave references 2-4, in which reported the use of carbon nanotubes (not just nanocomposites) in agriculture and medicine. These are modern articles (2021-2022) covering the specified topic.
  6. the main challenge of field is to elucidate the role of oxidative stress in the effects of CNTs on bacteria. This improves the understanding of the mechanisms by which CNTs act on bacteria and allows the selection of CNTs that have an antibacterial effect or support the formation of beneficial biofilms.
  7. We have tried to change some of the sentences in the Introduction to be more concise and clearer.
  8. We checked the abbreviations in the article and made adjustments. We gave the full name of the term with an abbreviation for the first time in the main text of the article, then we gave abbreviation only. We also gave the full name and abbreviation in the abstract.
  9. We formulated the problem at the end of the Introduction.
  10. In accordance with the remark, the goal was reformulated. «The objective of the work is to study the pro-oxidant and antioxidant effects of various types of CNTs on E. coli and to establish the effect of CNTs on the formation of ROS, biofilm formation, ATP content and morphological changes in planktonic and biofilm cells."
  11. We checked the manuscript and added spaces between number and units ( °C). In other cases, there are spaces between number and units in the original manuscript.
  12. Chemical list was added in section 2 (2.3).
  13. Comparison with literature data is presented in Discussion. Such experiments have not been described previously; this is the novelty of our study. We can only compare the effects of certain CNTs on bacteria, which have been described in previous articles. But this concerns other aspects (for example, mechanical disruption of cells).
  14. Comparison is given in the Discussion. Thus, the authors noted both the prooxidant and antioxidant effects of CNTs, and we note this in the Discussion and compare it with our data. Studies of the impact of SWCNT-ODA on bacteria have not been carried out previously.
  15. Data are presented as the average of three independent experiments. We added 2.9. Statistical analysis and Supplementary Materials. Adding the standard deviations on the graph makes it very difficult to read, so we made references to Supplementary Materials in Figure captions.

Figure 1. Statistical analysis is presented in the Supplementary Materials (Table S1)

Figure 2. …(Table S2)

Figure 3. …(Table S3)

Figure 5. …(Table S4)

  1. We added Challenges and future perspectives in Discussion.
  2. In Conclusion, we quantified the research findings and added highlights. Perspectives were added in Discussion.
  3. We rewrote some phrases, found a few mistakes. We would be very grateful to the reviewer for a direct indication of insufficiently clear phrases or errors in the text (with the number of line).
  4. Reference formatting was revised and found inaccuracies have been corrected.

Reviewer 2 Report

The main question is the influence of Carbon nanotubes on the oxidative stress response  and E. coli biofilm formation. It is very relevant and very interesting. The topic is very original. See the relevant literature. The originality is evident compared to the other references. The paper is well written. I have only suggested a minor correction.

The conclusions cover the essential points.

I found this paper very interesting. I only ask to make a single correction:

- line 417: decreases [34].

Author Response

We are very grateful to the Reviewer for the high opinion of our manuscript. Amendment mentioned has been made to the text.

Reviewer 3 Report

caborn nanotubes with different functionalisation were tested for bactericidal property. 

geometricsl factors of the used surfaces and their hydrophobic / hydrophilic nature can have very strong effect. AFM was used, it can provide deeper insight into substrates used (https://www.mdpi.com/2079-4991/10/5/873). 

insights into biocidal activity not well articulated nor compared with established mechanisms (see research of E Ivanova's group). 

there are mechanical bactericidal action of carbon nanotubes studied (High Aspect Ratio Nanostructures Kill Bacteria via Storage and Release of Mechanical Energy

DP Linklater, M De Volder, VA Baulin, M Werner, S Jessl, M Golozar, ...

ACS nano 12 (7), 6657-6667)

Is it relevant to the used experiments?

Author Response

Thank you for reviewing our manuscript.

We agree with the reviewer that the geometric factors of the surfaces and the hydrophobic/hydrophilic nature of the nanomaterials have a strong effect on bacteria. We discuss these points in our manuscript. Thus, the functionalization of CNTs leads to an increase in the hydrophilicity of the surface. We considered both intact (more hydrophobic) and functionalized (more hydrophilic) CNTs and showed that the prooxidant and antioxidant effects depend not on functionalization, but on whether SWCNTs or MWCNTs act on bacterial cells. We also emphasized this in the discussion: “Surface modification of CNTs affects their toxicity because the surface reactivity and their ability to aggregate with bacteria are changed. Hydrophobic unmodified CNTs are poorly dispersed, they agglomerate, and precipitation faster than those functionalized with hydrophilic groups [11].” We consider various mechanisms of CNT antibacterial activity in the Introduction: “The negative effect of CNTs on the cell can be associated with: 1) membrane disruption, or its oxidation; 2) oxidative stress caused by the formation of ROS; 3) toxicity of impurities; 4) bacterial agglomeration [11].” and in Discussion: “The antibacterial effect of CNTs depends on various factors. Among these are not only the properties of CNTs themselves (their diameter, length, functionalization, and others), but also the properties of the environment in which they operate."

In our manuscript we consider oxidative stress, not all mechanisms of nanomaterials antibacterial activity. This is reflected in the title of the manuscript.

With respect to the references mentioned, the work of Linklater et al. (DOI: 10.1021/acsnano.8b01665) is devoted to the study of the mechanisms of antibacterial activity of surfaces coated with nanostructures. We studied the effect of CNTs dispersed in the culture medium, on bacterial cells and the formation of biofilms, rather than CNTs in composite materials. The work of Linklater et al. (Nanomaterials 2020, 10, 873; doi:10.3390/nano10050873) is dedicated to "B-Si coated with TiO2 as a promising biocidal (anti-viral and bactericidal) surface, exhibiting strong oxidizing (electron removal) properties under UV light illumination that can be used to kill attaching microbes.” This is also irrelevant to our work. Using AFM, we studied the planktonic and biofilm cells of bacteria exposed to dispersed CNTs, rather than the interaction of bacterial cells with surfaces containing CNTs.

Reviewer 4 Report

The authors present an interesting and well written manuscript regarding the investigation of  the expression of oxyR and soxS oxidative stress genes in E. coli under the effect of different types of CNTS (single walled, multi and/or functionalized with  -COOH, -NH2,  groups etc)  The rationale for conducting this work is well-justified and the paper aligns nicely with the scopes of the journal.

For enhancing the quality of the manuscript:

I would recommend the authors to right a small paragraph in the introduction or incorporating in an existed one describing the antibacterial properties of CNTs and their use in biotechnology. Please add the following references:

1) Kouloumpis, A. et al. Germanane Monolayer Films as Antibacterial Coatings. ACS Applied Nano Materials 4, 2333-2338 (2021). https://doi.org:10.1021/acsanm.0c03149

2) Chalmpes, N. et al. Layer-by-Layer Assembly of Clay-Carbon Nanotube Hybrid Superstructures. ACS omega 2019, 4, 18100-18107, http://dx.doi.org/:10.1021/acsomega.9b01970.

3) Patila, M. et al. Chapter Twelve- Use of functionalized carbon nanotubes for the development of robust nanobiocatalysts. Methods in Enzymol.2020, 630, 263-301,https://doi.org/10.1016/bs.mie.2019.10.015.

Have you tried to zoom in more in the AFM in order to see if you can distinguish cleraly the presence of CNTs in the cells? If yes, please show them in the high quality AFM images you present in your paper, by putting an arrow. It is going to be very helpful to the readers.

From the cross sectional analysis of the AFM the corresponding values of thickness/size agrees with of what is mentioned in the bibliography and of the size of the commercial available materials that you used. Please see ref. 1 in the first comment.

Author Response

We thank the Reviewer for the high opinion of our manuscript. We have included the references [5,6,50] suggested by the Reviewer in the Introduction and in the Discussion.

Unfortunately, AFM makes it possible to obtain only surface profiles by scanning them with a cantilever. With this method, we will not be able to see nanotubes inside microbial cells. Nanotubes are visible on AFM images of biofilm cells with SWCNTs-COOH, on a substrate. Their size is consistent with the declared manufacturer. In our work, we used commercially available carbon nanotubes with known characteristics.

Round 2

Reviewer 1 Report

Accepted in the present form. Author revised it very well. 

Reviewer 3 Report

mechanical contact of any nano-object and bio-cells in liquid is affected by Brownian motion regardless of dispersed nanomaterials or textured surfaces.  what is considered "irrelevant" might be very different